# Diffusion models enable zero-shot pose estimation for lower-limb prosthetic users

**Tianxun Zhou**[1☯]**, Muhammad Nur Shahril Iskandar**[2☯]**, Keng-Hwee Chiam**[1]*

**1** Bioinformatics Institute, Agency for Science, Technology and Research, Singapore, Singapore, **2** Physical Education and Sports Science Academic Group, National Institute of Education, Nanyang Technological University, Singapore, Singapore

☯ These authors contributed equally to this work.
* chiamkh@bii.a-star.edu.sg

**Data availability statement:** All the raw videos used in this study of individuals with unilateral lower-limb prosthetic walking are available for download from the publicly available Mission Gait YouTube Channel at https://www.youtube.com/@MissionGait/videos.

**Funding:** The author(s) received no specific funding for this work.

**Competing interests:** The authors have declared that no competing interests exist.

## Abstract

Quantitative gait analysis is important for assessing and rehabilitating lower-limb prosthetic users, but markerless motion capture has been challenging for this population due to the difficulty in detecting prosthetic joints using models trained primarily on able-bodied individuals. This study proposes a zero-shot method leveraging generative diffusion models to transform prosthetic limb images into able-bodied representations that standard pose estimation models can detect, eliminating the need for additional data collection or model retraining. Videos of unilateral transfemoral and transtibial amputees walking were obtained publicly from YouTube. For each video frame, an edge map was generated and used as input to a ControlNet diffusion model, generating a synthetic image resembling an able-bodied person while preserving the person's original pose. These synthetic images were then passed through OpenPose. The zero-shot approach achieved substantial reductions in keypoint coordinate errors of 37% for transtibial and 76% for transfemoral prosthetic limbs compared to OpenPose on the original videos. The method enabled the identification and quantification of key gait deviations such as reduced knee flexion and altered kinematics timing between prosthetic and intact limbs. While the results demonstrate the feasibility of markerless gait analysis for lower-limb prosthetic users, the study's findings are based on a limited dataset of publicly available videos, and caution should be exercised in generalizing the results to broader populations due to the varying nature of prosthetic designs. Nonetheless, this approach has the potential to facilitate personalized rehabilitation using standard consumer cameras and existing pose estimation models.

## Author summary

The application of 2D markerless gait analysis has garnered increasing interest and application within clinical settings. However, its effectiveness in the realm of lower-limb

amputees has remained less than optimal due to a lack of large image datasets of lower-limb prosthetic users to train models with. In response, we introduce an innovative zero-shot method employing image generation diffusion models to achieve marker-less pose estimation for lower-limb prosthetic users without the need for additional data collection and training, presenting a promising solution to gait analysis for this specific population. Our approach demonstrates an enhancement in the detection of key points on prosthetic limbs over existing methods, and enables clinicians to gain invaluable insights into the kinematics of lower-limb amputees across the gait cycle. The outcomes obtained not only serve as a proof-of-concept for the feasibility of this zero-shot approach but also underscore its potential in advancing rehabilitation through gait analysis for this unique population.

## Introduction

Human gait, especially in lower limb amputees, is a complex motor task requiring coordinated movement of various body segments. Extensive research has established that individuals in these populations frequently exhibit distinct gait abnormalities and physiological challenges. For example, it is known that gait asymmetry [1,2] is prevalent, where the prosthetic and intact limbs display different walking patterns. The presence of gait asymmetry may contribute to the development of degenerative joint disease [3]. Previous studies on physiological differences have also revealed that during walking, above-knee amputations lead to an increase in oxygen consumption by approximately 49% [4], in addition to an approximately 65% higher energy expenditure when compared to individuals without amputations [5]. These issues could hinder mobility and functionality, ultimately diminishing the quality of life [6]. Therefore, clinicians are often keen to understand the gait patterns of lower limb amputees both in general and at an individual level, in order to improve quality of life of prosthetic users through better rehabilitation monitoring, providing better fitted prosthetic, reducing gait abnormalities and more.

Gait analysis offers a systematic approach to identify pathological gait patterns associated with neurological [7], musculoskeletal [8], and other disorders by collecting various gait parameters data [9,10]. This allows clinicians to identify deviations from normal gait patterns and tailor interventions to address the specific issue. Gait analysis can be broadly classified as either observational [11] or quantitative.

Quantitative gait assessments have been facilitated by technologies such as instrumented motion analysis [12], force platforms, electromyography [1], and wearable sensors. These tools allow clinicians to systematically quantify a spectrum of gait parameters, enabling them to effectively monitor progress throughout the rehabilitation journey. For instance, Sjödahl et al. (2022) [13] used force-plates and motion-capture systems to identify increased hip flexion at the beginning of the stance phase on the intact limb, offering timely feedback to the patient. Furthermore, intricate spatiotemporal variables such as step length and cadence could be scrutinized in meticulous detail. The study revealed a reduction in variability on the prosthetic limb, indicative of a more stabilized and symmetrical gait pattern post treatment. Such nuanced, comprehensive, and timely analysis would not be possible if clinicians were tasked with manually tracking each individual step.

Traditionally, obtaining quantitative measurements demands specialized, often costly equipment, which confines data collection to controlled laboratory environments, subsequently limiting the broader applicability of findings [14,15]. In recent years, an alternative has emerged through the utilization of high-speed video recording on commercial video

cameras. Advances in deep learning have enabled rapid progress in human pose estimation from videos [16–19]. Pose estimation software such as OpenPose [20] are able to estimate the position of keypoints such as joints on images of humans without the need for markers. With multiple cameras, it is possible to triangulate the keypoints to obtain 3D coordinates [21]. Such markerless pose estimation techniques may potentially be the key to cost-effective systems for automated clinical gait analysis. Several studies have examined the accuracy of available pose estimation methods specifically for gait analysis applications [10,22–24].

Current markerless pose estimation models face challenges in accurately localizing joints on prosthetic limbs [25]. This limitation stems from their training on datasets primarily featuring able-bodied individuals, resulting in poor generalization to unseen prosthetic limbs. Custom models tailored for prosthetic keypoint identification have been proposed, yet their applicability across diverse settings is constrained by the wide array of prosthetic appearances and limited training data [25,26]. Achieving robust generalization across varied settings necessitates training on extensive datasets akin to those used in general human pose estimation, like the COCO dataset underlying OpenPose. However, the resource-intensive manual labeling of such datasets presents a formidable barrier in terms of cost and time and would likely be beyond the reach of rehabilitation labs.

This paper introduces a novel zero-shot pose estimation method leveraging existing pre-trained image generative diffusion models and pose estimation frameworks to achieve accurate pose estimation for lower limb prosthetic users. Recent advances in image generation artificial intelligence, notably with denoising diffusion models, have enabled the generation of remarkably realistic images conditioned on diverse inputs, including text descriptions and sample images [27]. Leveraging the capabilities of these image generation models, we transform prosthetic limbs in images into representations resembling able-bodied limbs while preserving their position and shape. This technique empowers existing pose estimation models to make accurate keypoint estimations on lower-limb prosthetics without further training.

## Methods

We analyzed publicly accessible videos of individuals with unilateral lower-limb prosthetic walking from video sharing website uploaded by Mission Gait YouTube Channel. For each video, an image-generative model, ControlNet, was utilized to generate synthetic images corresponding to every frame of the video. Subsequently, OpenPose was applied to these newly synthesized images, allowing for the extraction of anatomical keypoints. Inverse kinematics was performed based on the 2D coordinates generated. We then evaluated the performance based on a custom model created using DeepLabCut (DLC). Gardiner et al. (2016) [28] have shown that using publicly available videos produces results that are comparable with published data from controlled laboratory studies.

### Dataset

The video search was conducted by a single researcher with terms including: gait, walking, amputee, transtibial, and transfemoral. A total of 16 videos containing two subjects with different amputation levels; a female transfemoral and a male transtibial amputee, were downloaded. Essentially, each subject has two sets of four videos (S1 Fig), recorded at different time points (i.e. Transfemoral 1 and 2) and different camera positions (i.e., anterior, posterior, left and right). Both subjects are amputees on the right side. For the transfemoral subject, the same prosthesis was used across all recordings. In contrast, the transtibial subject used

two different types of prostheses: a running blade in Transtibial 1 and a prosthesis resembling a conventional lower limb in Transtibial 2. As the source of videos was publicly available, the etiology of amputation, time since amputation, time between the two recordings, and the exact type of prosthetic used were unknown. The videos were downloaded at a resolution of 1280 x 720 and 30 frames per second. Subsequently, each video was trimmed to 6 seconds (180 frames) as the initial videos included the subject changing walking direction, before following the zero-shot pose estimation pipeline.

## Zero-shot pose estimation pipeline

The pipeline for zero-shot pose estimation is as follows (Fig 1). Given a video, for each frame, we first apply edge detection using Canny edge detection to generate an edge map. The edge map is then used as the conditional control into a text-to-image generative diffusion model. In this case, we use ControlNet proposed by [27] which applies conditional controls using inputs such as edge maps, segmentation maps, and keypoints to pretrained large diffusion models to support additional input conditions other than prompt text. The generated image

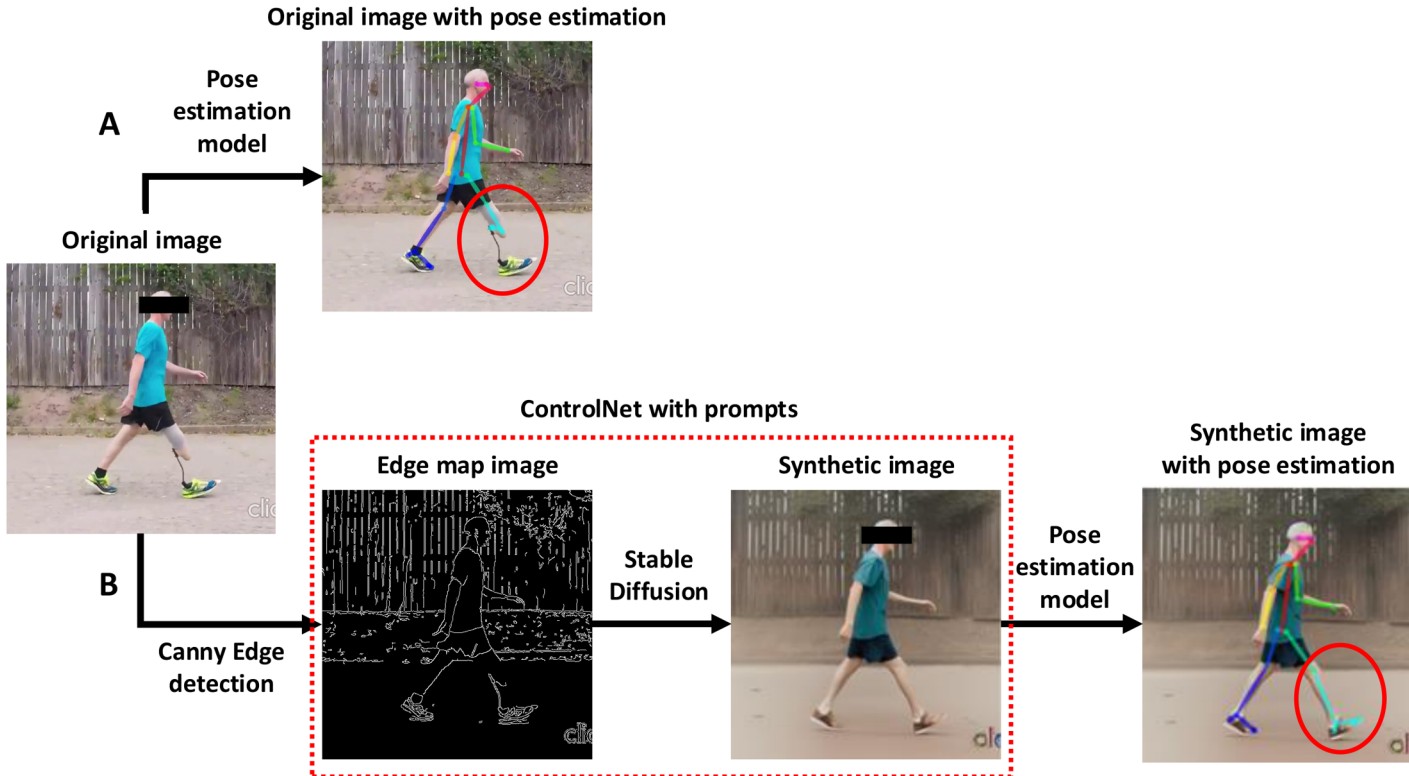

**Fig 1. Workflow of the proposed zero-shot method.** (A) The standard workflow for applying pose estimation (e.g., OpenPose) directly on the original image. In this example, OpenPose fails to accurately identify keypoints on the prosthetic limb, as highlighted by the red circle. (B) The proposed zero-shot workflow leverages ControlNet. The original image is processed using the Canny edge detector to generate an edge map, which is then passed through Stable Diffusion to create a synthetic image. This synthetic image is subsequently used for pose estimation. In this case, improvements in pose estimation accuracy, particularly in the identification of lower-limb keypoints, are evident, as highlighted in the red circle.

by the diffusion model conforms to the edge map to a large extent but transforms the prosthetic limbs into able-bodied limbs. The generated image is then passed to a standard pose estimation model, specifically, OpenPose, trained on large human pose estimation datasets to output the pixel locations of body keypoints. The frames were rescaled to 512 x 512 to adhere to the default ControlNet configuration.

## Edge map conditioned diffusion model

Image generative diffusion models are currently state-of-the-art models for generating synthetic images. They are inspired by the diffusion process where molecules move from high to low-concentration areas, leading to a homogenization of the distribution over time. In image generation, the diffusion process is applied to the image pixels, where random noise is added over a number of time steps until the image is indistinguishable from random Gaussian noise.

The diffusion process is typically implemented using a sequence of invertible transformations, such as Gaussian diffusion or Langevin dynamics. During training, the model learns to reverse the diffusion process given the time step and the noisy image at that time step. During inference, new images can be generated by sampling from random Gaussian noise and applying the learnt reverse diffusion step-wise to obtain a clean generated image.

To generate images conditioned upon other features such as class, text description, or sample image, the conditioning information is provided to the diffusion model as an additional input. The conditioning information is typically encoded into a fixed-length vector representation using a separate neural network, such as a text encoder, which is then used as an input to the diffusion model.

During training, the diffusion model is trained to maximize the log-likelihood of the observed data given the conditioning information. During inference, the conditioned diffusion model is used to generate images by sampling from the diffusion process given the conditioning information. The conditioning information is used to initialize the diffusion process and to guide the sampling process at each time step.

Overall, conditioning diffusion models for image generation involves modifying the training and inference procedures to incorporate the conditioning information, and training a separate network to encode the conditioning information into a fixed-length vector that can be used as input to the diffusion model.

ControlNet is a neural network architecture proposed by [27] that applies task-specific controls to large image diffusion models such as Stable Diffusion. ControlNet does so by changing the inputs of diffusion network blocks. ControlNet makes a copy of the network block and applies a 1x1 convolution layer before and after the block. The corresponding diffusion model network block weights are frozen during training to avoid overfitting to the smaller conditional training set and preserve the model's pre-trained high-quality image generation capabilities. The control condition is fed to the first 1x1 convolution layer and the output of that layer is added to the original input of the network block. This modified input is then passed through the network block and the subsequent 1x1 convolution layer. The resulting output is added to the original output of the network block [27].

On the Stable Diffusion model, which follows a U-Net architecture, ControlNet is implemented to control each level of the U-Net. It creates trainable copies of the 12 encoding blocks and 1 middle block of Stable Diffusion. The outputs of the ControlNet are added to the 12 skip-connections and 1 middle block of the U-Net. We refer interested readers to the detailed explanation in the original paper.

ControlNets have been trained to perform tasks based on various conditions, this includes generating images that are controlled by Canny edges, Hough lines, HED edges, human pose keypoints, segmentation maps, depth maps and more.

Edge maps were generated using the Canny edge detector, which works by identifying areas of rapid intensity change, thus highlighting the boundaries of objects in the image [29]. It starts by applying a Gaussian filter to smooth the image, which helps reduce noise and makes the edges clearer. Next, it calculates the intensity gradients to identify areas where there is a rapid change in brightness, indicating potential edges. To determine which edges are significant, the detector uses two threshold values: a high threshold to mark strong edges and a low threshold to include weaker edges that are connected to these strong ones. This approach ensures that important edges are preserved while minimizing noise, resulting in a clear edge map that highlights the boundaries of objects in the image. These edge maps subsequently serve as the conditioning information for the diffusion model.

Specifically relevant to this application, the Canny edge conditioned ControlNet were trained with edge maps that were obtained by processing 3 million images (with captions) using the Canny edge detector. We used the default configuration such as an annotator resolution of 512, a canny low threshold of 100, and a canny high threshold of 200. The prompts used to generate images for this study were, "an able-body person walking, intact lower limbs, 2 legs, full-body portrait, realistic", while the negative prompts were, "cyborg, amputee, pan-futurism". The code for ControlNet is created by [27] and available at https://github.com/lllyasviel/ControlNet.

## Pretrained pose estimation model

Human pose estimation models aim to estimate the pose of humans in 2D images, typically by detecting body keypoints. Deep learning has led to significant advances in human pose estimation, and models such as OpenPose and AlphaPose, which have been trained on large datasets (COCO human keypoints dataset), demonstrated good performance on various benchmarks. Several studies also investigated the accuracy of these pretrained pose estimation models for clinical applications including gait analysis. Here we summarize OpenPose in more detail, which is the widely used model in pose estimation for clinical and biomechanics studies [24,30–32].

The OpenPose model uses a multi-stage convolutional neural network (CNN) architecture to process images and estimate human poses [20]. The keypoint detection is performed in two stages: the first stage generates a coarse heatmap of body keypoints and parts affinity field, and the second stage refines the heatmap to produce a more accurate estimate of the keypoint locations. The model then uses the key point heatmap and parts affinity field to link up the keypoints into the pose skeleton. The model can detect various body keypoints, including those of the head, neck, shoulders, elbows, wrists, hips, knees, and ankles. OpenPose is designed to work in real-time and is capable of estimating human poses in images or videos with multiple people, even in complex scenes with occlusions and overlapping body parts. It is also able to estimate the pose of people in different orientations and viewpoints.

Another pose estimation model used in this study was DeepLabCut (DLC) version 2.3.0. DLC allows users to label and train with their own key points. Its architecture consists of a residual neural network (ResNet-50) with deep convolutional and deconvolutional neural network layers to predict the keypoints using feature detection. We train a DLC model using videos from the transtibial amputee. Ground truth labels for 8 lower-body keypoints (left and right of hip, knee, ankle, and toes) were obtained by manual annotation of every frame for one set of the transtibial videos. The prosthesis-side keypoints were aligned with intact-side

keypoints as closely as possible when anatomical landmarks were inaccessible or absent, similar to other studies on transtibial [33] and transfemoral [34] amputees. Using the labels of these 4 videos as ground truth, we evaluated the accuracy of the custom model (using DLC) trained on 20 labelled frames until the training loss had plateaued.

The data, expressed in mean (and standard deviation in parenthesis), are reported in pixels due to unknown measurements and the lack of a calibration procedure in the video. Results from our preliminary study were deemed satisfactory which showed that the custom model had a mean absolute error (MAE) of 7.89 ± 3.11 pixels for the coordinates (S5 Table) and 1.61° ± 1. 19° for kinematics when evaluated on one set of transtibial videos (S6 Table). Therefore, a custom model following the same steps was taken to obtain the ground truth coordinates for the other 12 videos. The resulting keypoints were visually examined frame by frame to ensure accuracy before we effectively treated them as manual labels.

## Error quantification on keypoints coordinates

In certain frames, the visibility of keypoints may be occluded leading to lower likelihood scores given by OpenPose. Consequently, frames with a likelihood below 0.50 were removed (S7 Table). Cubic spline interpolation was applied to address the resulting gaps in the data, and a low-pass Butterworth filter (4th order, 6 Hz) was subsequently applied to attenuate high-frequency noise. These procedures have also been used in previous studies [22,35].

## Error quantification on joint kinematics

Assuming that the left and right camera views are positioned perpendicularly relative to the sagittal plane, lower-limb joint angles of the limb closest to the camera can be calculated using the keypoints coordinates via inverse kinematics [10,22–24]. The hip angle was calculated involving the keypoint vectors of the hip, knee and a relative vertical line to the hip. Similarly, the knee angle was computed by using the keypoint vectors of the hip, knee, and ankle. In both cases, a positive value indicates flexion while a negative value indicates extension. The ankle angle was calculated using the keypoints vectors of the knee, ankle, and toe. In addition, the ankle angle is defined by the foot with respect to a 90° line to the tibia. The kinematic data for the entire time-series was initially calculated, followed by time normalization from 0-100% of the gait cycles. Gait cycles were defined as consecutive occurrences of heel strikes by the same foot, which were manually identified. There were at least 4 gait cycles observed for all videos. Another gait parameter of interest among clinicians is the lateral trunk lean, measured from the anterior camera view. The lateral trunk lean angle was calculated using the keypoint vectors between the mid-hip, neck, and vertical. This gait parameter was only calculated for the OpenPose and zero-shot methods, as no torso coordinates were labelled in DLC to obtain ground truth. To ensure consistency with existing literature, we calculated the lateral trunk lean angle in both abled and prosthetic limb gait cycles.

To quantify the error of pose estimation, we use the MAE which is the mean Euclidean distance between ground truth and predicted keypoints. Both coordinates and kinematics data were quantified using MAE. There are several previous works in the literature that quantify the accuracy of markerless pose estimation models such as OpenPose for the purpose of gait analysis. S4 Table summarizes existing data reported on the accuracy of clinical gait applications of markerless pose estimation on able-bodied individuals. Original values reported are used if provided in the papers, otherwise, the values were estimated by reading off from figures.

## Results

The experimental findings demonstrate significant improvement in pose estimation accuracy achieved by our zero-shot method compared to the established pose estimation software, OpenPose. This is a crucial improvement needed for practical gait analysis, such as comparing joint angles through the gait cycle, which is of interest to clinicians. Additionally, we also discern and quantify variations in pose estimation performance across different prosthetic types.

### Obvious misidentification in OpenPose resolved by zero-shot method

Our findings reveal that when OpenPose is directly applied to the original image, a substantial proportion of frames exhibit either an inability to identify or a conspicuous misidentification of lower-limb keypoints (Fig 2). S1 Table provides a detailed breakdown of these occurrences. Specifically, transfemoral images display a higher incidence of such challenges compared to their transtibial counterparts. Notably, our zero-shot method demonstrates remarkable efficacy in mitigating the number of affected frames across all camera perspectives. It is also important to note that the majority of observed keypoint challenges are concentrated in the sagittal camera view, particularly when the prosthetic limb is in closer proximity to the camera.

### Substantial improvement in accuracy of quantitative measures using zero-shot method

The OpenPose method exhibited coordinate errors that were about twice as large (Table 1) on the prosthetic limb (right) compared to the intact limb (left). A detailed breakdown of the individual keypoints coordinates errors can be found in S2 Table.

The zero-shot method consistently outperformed the OpenPose method in estimating keypoints on the prosthetic limb by a large margin for joint coordinates. The mean absolute error (MAE) measured for joint coordinates decreases by 37% for transtibial prosthetic limbs and 76% for transfemoral prosthetic limbs. Even though OpenPose method yielded slightly lower coordinate errors on the intact limb, the differences observed were relatively minor, with a mean difference of less than 2 pixels.

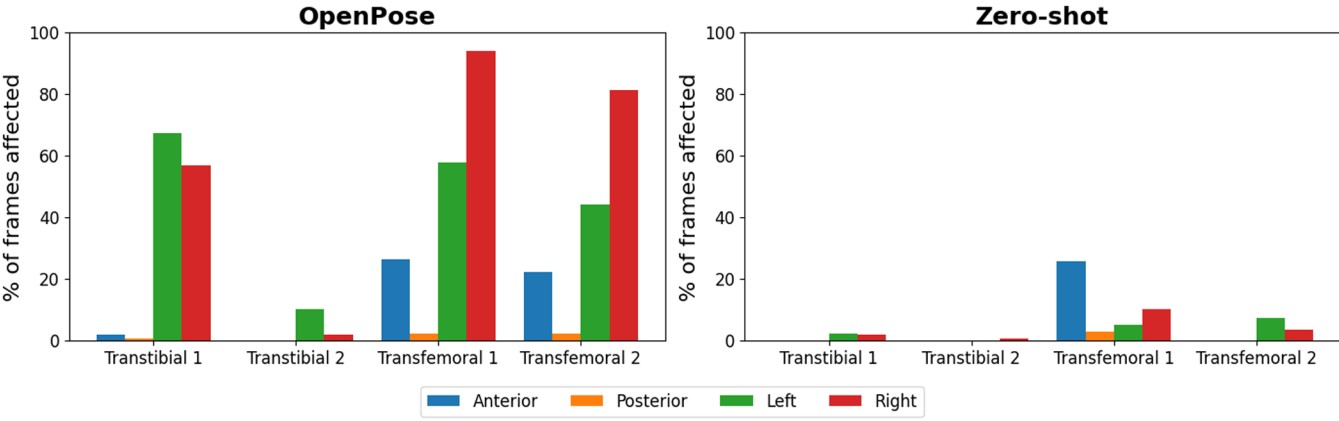

**Fig 2. The percentage of failures in each video.** Values are expressed as percentages based on a total of 180 frames. Each graph shows the percentage of frames where lower-limb keypoints were not identified fully. Each colour represents the different camera positions.

**Table 1. The overall mean absolute error (MAE) and standard deviations (SD), in parentheses of keypoint coordinates error (MAE) after processing using the default OpenPose and zero-shot method.**

| Metrics (Units) | Amputation type | Leg | OpenPose (MAE) | Zero-shot (MAE) |
|---|---|---|---|---|
| Joint Coordinates (pixels) | Transfemoral | Prosthetic | >100 (>100) | **23.7 (35.61)** |
| | | Intact | **10.22 (5.36)** | 11.96 (7.58) |
| | Transtibial | Prosthetic | 24.07 (15.74) | **15.18 (6.66)** |
| | | Intact | **12.01 (3.54)** | 12.27 (7.60) |
| Joint Kinematics (°) | Transfemoral | Prosthetic | 29.64 (26.07) | **6.66 (6.50)** |
| | | Intact | 5.71 (4.77) | **5.23 (5.10)** |
| | Transtibial | Prosthetic | 10.23 (9.08) | **6.32 (4.49)** |
| | | Intact | **2.90 (2.11)** | 3.25 (2.54) |

For all videos, the right leg is the prosthetic limb while the left leg is the intact limb. Bold font indicates the lower value between OpenPose and the Zero-shot method.

Similarly for joint kinematics, the errors against ground truth using the zero-shot method consistently outperformed the OpenPose method on the prosthetic limb by a large margin. The improvement is particularly noteworthy for ankle angles and transfemoral knee angles. The comparison of joint kinematics is shown in Fig 3.

Although the open-source videos lacked explicit calibration or measurement information, an estimate of the subject's height enabled us to approximate the metric error. Assuming a subject height of approximately 180 cm, we deduced that 1 pixel corresponded to approximately 0.3 cm. These assumptions enabled us to draw comparisons with previous findings. Our results for OpenPose on subjects with intact limbs are consistent with existing literature (S4 Table). Clinicians aiming to replicate our study may consider incorporating calibration steps at the beginning of the video recording. This would allow them to gather additional gait parameters, such as stride length and step width, which were not analyzed in our study due to the utilization of open-source videos.

## Comparison of performance between different prosthetic types

The two main types of lower-limb prosthetics, transfemoral and transtibial differ in appearance and in characteristics of gait. The performance of pose estimation and gait analysis on transfemoral and transtibial amputees are compared.

As observed in Table 1, when comparing transfemoral and transtibial amputees, the pose estimation of the prosthetic limb exhibited poorer performance in the transfemoral amputee. Even with the application of the zero-shot method, the coordinate errors on the prosthetic limb were twice as substantial in the transfemoral amputee. Whereas in the transtibial amputee, the zero-shot method was able to reduce the coordinate errors on the prosthetic limb to a comparable level as the intact limb.

The hip angle curve exhibited satisfactory results for both transtibial (Fig 3) and transfemoral (Fig 4) amputees when employing both the OpenPose and zero-shot methods. Particularly, the zero-shot method displayed the ability to reduce the MAE on the prosthetic limb for the transfemoral condition by about 3 degrees.

In the knee angle, the OpenPose method was able to replicate the kinematic curvature throughout the gait cycle in the transtibial amputee (Fig 3) but less so in the transfemoral amputee (Fig 4). Thereafter, the application of the zero-shot method resulted in an improvement in replicating the knee angle kinematic throughout the gait cycle for the transfemoral amputee (Fig 4).

In comparison to the hip and knee angles measured using OpenPose, the ankle angle exhibited higher error values. This observation is further supported by the higher error

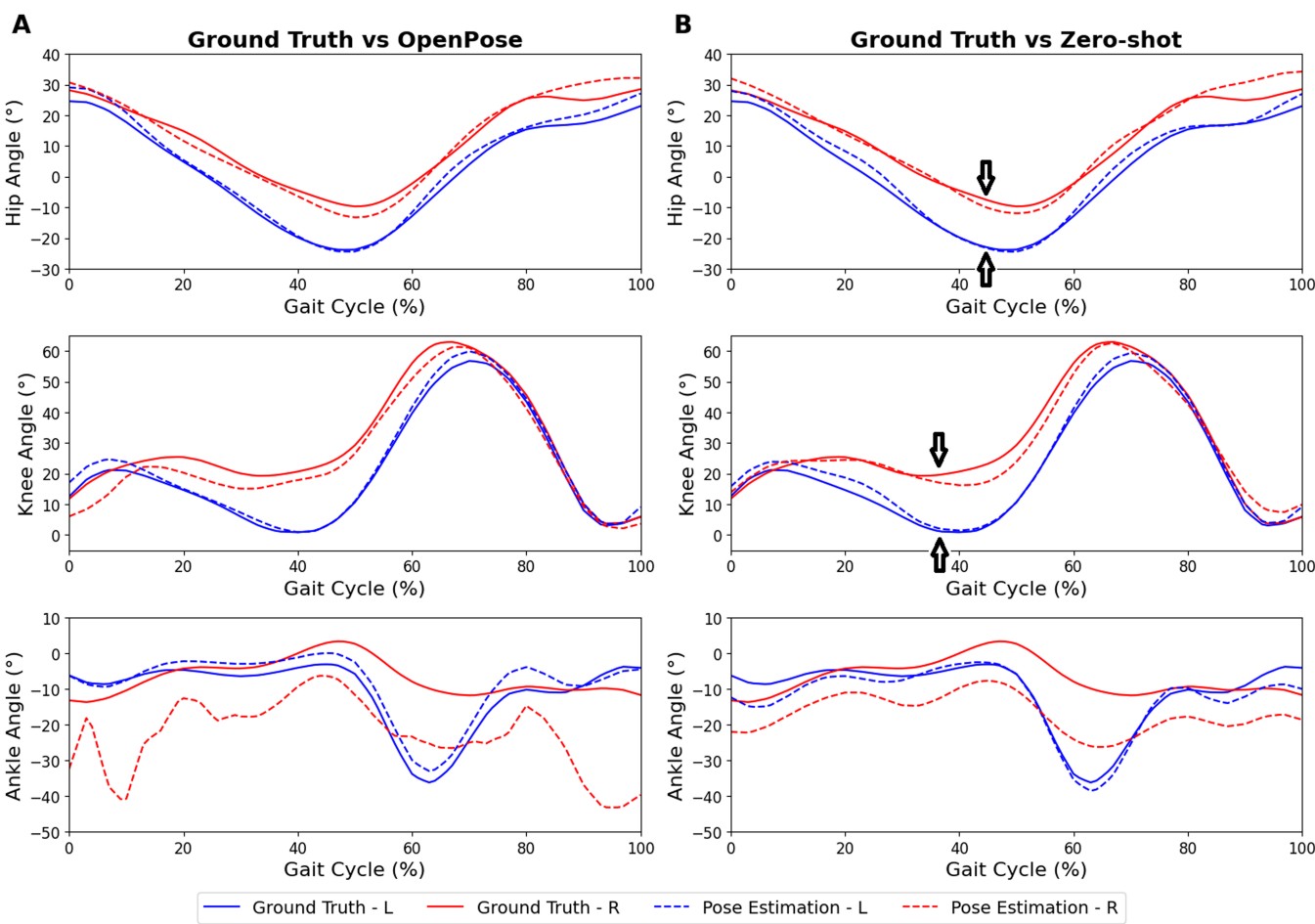

**Fig 3. Normalized kinematics results for transtibial.** Top and bottom rows represent hip, knee, and ankle kinematics, respectively, on the left (blue; intact) and right (red; prosthetic) limb. (A) Kinematics comparing the ground truth (solid line) and OpenPose (dotted line). (B) Kinematics comparing the ground truth (solid line) and zero-shot method (dotted line). Positive hip and knee joint angles indicate flexion, negative angles indicate extension. Positive ankle joint angles indicate dorsiflexion, negative angles indicate plantarflexion. Some key differences in kinematics between the prosthetic and intact limbs are marked with arrows

in ankle keypoint coordinate (S3 Table). When the zero-shot method was applied, a slight improvement in replicating the ankle kinematic curve for the transtibial amputee was observed, but no improvement was evident for the transfemoral amputee.

## Gait anomalies in prosthetic limb revealed by zero-shot method

Previous studies have indicated significant kinematic differences between prosthetic and intact limbs in lower-limb amputees [36]. Notably, the kinematic gait cycle observed in our study exhibited a similar pattern to those reported in previous studies [2,37].

For instance, as seen in Fig 3, our zero-shot approach revealed that the hip extension of the transtibial amputee was reduced. Additionally, within the 10-50% range of the gait cycle, a decreased range of motion was observed in the knee angle. In the case of the transfemoral amputee as seen in Fig 4, a noticeable phase shift in both hip and knee angles was observed, indicating a leftward displacement in the kinematic patterns throughout the gait cycle when compared to the intact limb's kinematics. In comparison, OpenPose fails to capture some of

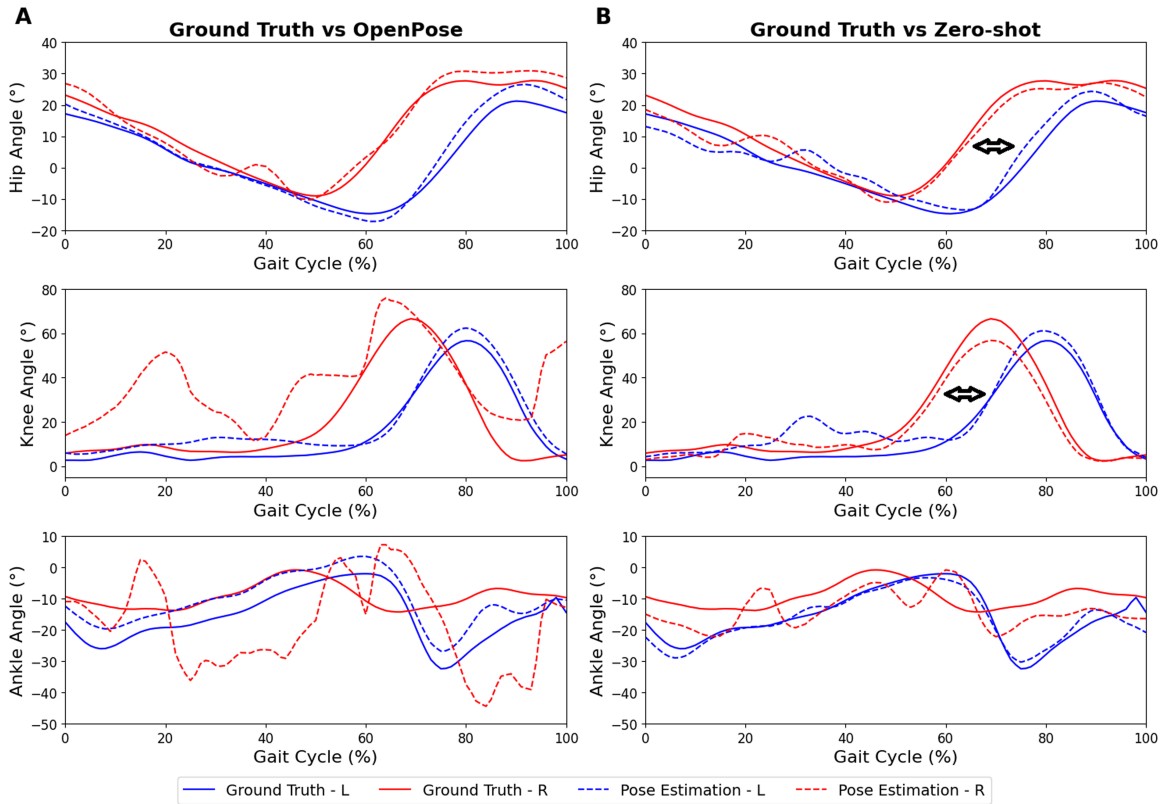

**Fig 4. Normalized kinematics results for transfemoral.** Top and bottom rows represent hip, knee, and ankle kinematics, respectively, on the left (blue; intact) and right (red; prosthetic) limb. (A) Kinematics comparing the ground truth (solid line) and OpenPose (dotted line). (B) Kinematics comparing the ground truth (solid line) and zero-shot method (dotted line). Positive hip and knee joint angles indicate flexion, negative angles indicate extension. Positive ankle joint angles indicate dorsiflexion, negative angles indicate plantarflexion. Some key differences in kinematics between the prosthetic and intact limbs are marked by arrows

these findings, especially in the transfemoral case due to much more inaccurate joint angle kinematics. This demonstrates the effectiveness of the zero-shot method over OpenPose for identifying and quantifying key anomalies during gait analysis and provides the means for clinicians to uncover and measure key gait markers during rehabilitation. Similarly, the lateral trunk lean angle was analyzed from the anterior camera view. Although error quantification was not possible due to the absence of ground truth labels, both amputees exhibited a lateral trunk lean toward the prosthetic limb during the stance phase (Fig 5). The kinematic curves generated by OpenPose and the zero-shot method were also quite similar.

## Discussion

To our knowledge, this is the first work that achieves zero-shot pose estimation for lower-limb prosthetic users. The contributions of this paper are twofold. First, we provide quantification of the errors made by one of the most widely used pre-trained pose estimation software, OpenPose, on lower-limb prosthetic users. Second, we demonstrate a working method for accurate zero-shot pose estimation on lower-limb prosthetic users without the need for any data collection, labelling or training. Our zero-shot method achieved 37% to 76% decrease in error for pose estimation on transtibial and transfemoral prosthetic limbs respectively. This work serve to demonstrate the feasibility of the zero-shot approach which is a step towards

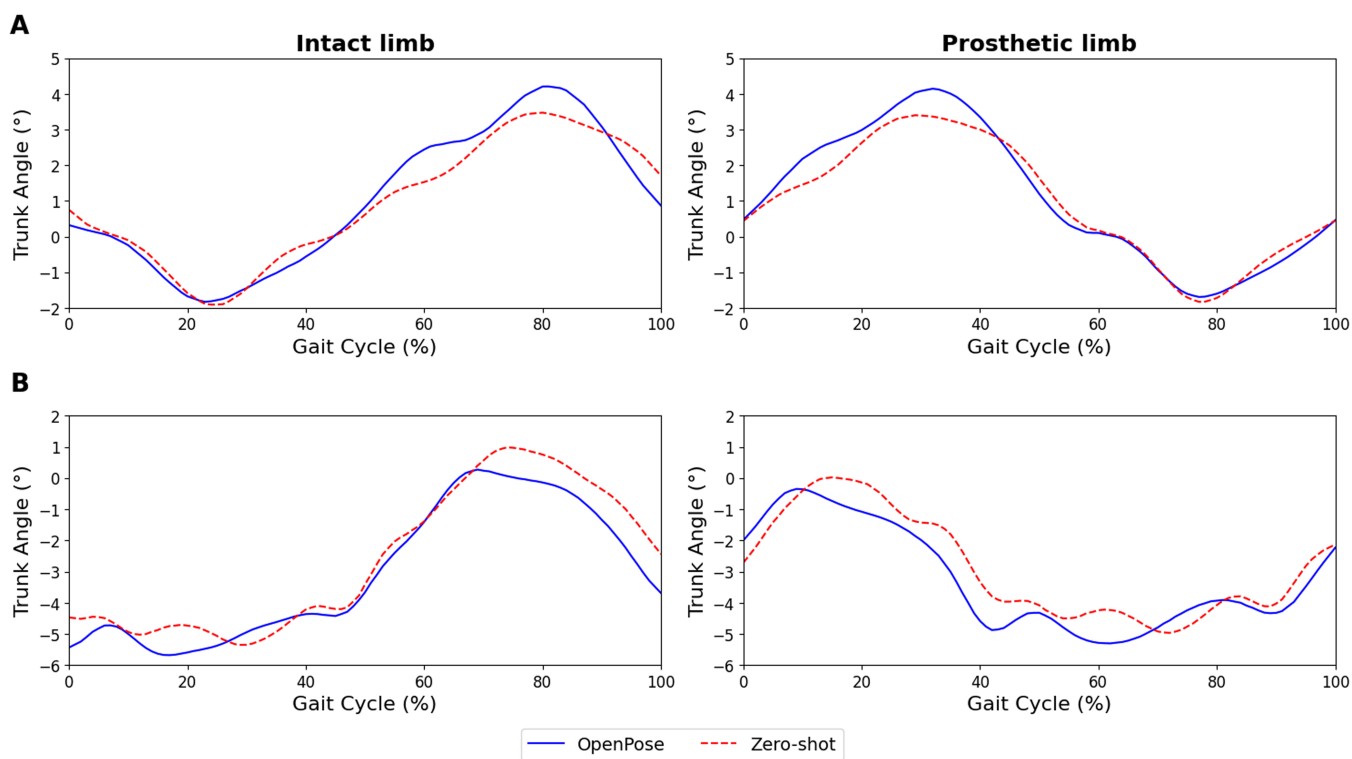

**Fig 5. Normalized lateral trunk lean angle.** (A) Transtibial and (B) Transfemoral amputee. Each colour represents the pose estimation used: OpenPose (blue) and the zero-shot method (red). Positive angles indicate trunk lean to the ipsilateral side, and negative angles indicate trunk lean to the contralateral side.

advancing personalized rehabilitation and improving quality of life outcomes through gait analysis for the lower-limb amputee population. By improving gait analysis for the lower-limb amputee population, our approach has the potential to enhance the development of custom rehabilitation strategies, ultimately improving mobility and overall quality of life outcomes for prosthetic users.

### Improved pose estimation accuracy enables quantitative analysis of gait biomechanics in lower-limb amputees

Gait analysis for lower-limb prosthetic users without the need for specialized equipment has the potential to transform rehabilitation for this population, by providing automated, consistent, and quantitative assessment of the presence and severity of pathological gait. This opens up the possibility of rapid clinical feedback, reliable longitudinal tracking of progress, objective comparison between rehabilitation strategies, and more. The method proposed in this work is a step towards this goal through accurate pose estimation from markerless videos acquired from low-cost commercial cameras.

The results obtained in our investigation demonstrate the effectiveness of the zero-shot method in reducing both the coordinates and kinematics errors when compared to applying OpenPose on the original images. Although the ankle kinematics exhibited a relatively higher error value, the hip and knee kinematics generated using the zero-shot method remain valuable for clinicians. Such kinematic information enables clinicians to discern disparities in kinematic patterns between intact and prosthetic limbs, thereby facilitating the planning

and monitoring of individualized rehabilitation programs. Presently, it is evident that Open-Pose encounters difficulties in accurately detecting images featuring prosthetic limbs, with higher error rates observed for prosthetic limbs compared to intact limbs (Fig 2). This observation aligns with [25], who similarly reported challenges when utilizing OpenPose for lower-limb amputees, consequently restricting the applicability of markerless motion capture in this population.

In the anterior camera view, lateral trunk lean is a common compensatory strategy observed among lower-limb amputees [38–41]. Consistent with findings in existing literature [39,41], our analysis also revealed that both transtibial and transfemoral subjects leaned toward the prosthetic limb during the stance phase. When comparing the kinematic curves generated by OpenPose and our zero-shot method, we found little differences between them. This may suggest that current pose estimation methods may already offer sufficient accuracy for frontal-plane analysis. This is unsurprising, as the calculation of lateral trunk lean angles does not account for the presence of the prosthetic limb. Thus, altering a prosthetic limb to resemble an able-bodied limb in our zero-shot method has minimal impact on the resulting kinematic outcomes. Practitioners may opt to use pose estimation algorithms directly to quantify lateral trunk lean angles, though the accuracy of these measurements remains to be thoroughly validated.

Our findings demonstrate that the implementation of the zero-shot method leads to enhanced keypoint detection by greatly reducing the number of problematic frames, subsequently reducing coordinate and kinematic errors. This improvement has practical implications for gait analysis, particularly in deriving kinematic angles throughout the gait cycle. This information is crucial for clinicians to comprehend the gait symmetry of individuals. From a clinical standpoint, the analysis of kinematics throughout the gait cycle provides the ability to discern asymmetrical walking patterns between the prosthetic and intact limb. When compared to OpenPose, we have shown the zero-shot method to be more effective in identifying and quantifying significant anomalies during gait analysis. By incorporating the identified anomalies and utilizing the insights gained from the zero-shot method, the clinician can design an individualized rehabilitation regimen that addresses specific gait deviations and promotes the restoration of balanced walking patterns. This personalized approach enhances the potential for successful rehabilitation outcomes in the lower-limb amputees' population.

## Limitation, possible solutions, and future work

The performance of our zero-shot method is inherently dependent on the quality of the input images and the resulting edge maps. From our experience, the Canny edge detector's effectiveness can be influenced by variations in lighting, texture, and noise levels in the original images. This variability can lead to several issues, such as low-contrast regions failing to produce well-defined edges, resulting in incomplete edge maps. Moreover, details like overlapping limbs, might not be adequately captured, causing ambiguities in the edge map. This limitation is particularly evident in our observation of increased swapping of lower-limb keypoints in the sagittal plane ( S1 Table). For instance, if the left leg is closer to the camera, the left leg will naturally occlude parts of the right leg that are directly behind when viewing from the camera. Hence, when the edge detector fails to identify the outline of the leg in the region where the two legs overlap, it will result in an ambiguous edge map. This ambiguity may lead to inaccuracies in the information provided to the diffusion model, causing a swap between the two legs in the synthetic image. Fig 6A provides an illustrative example of the observed phenomenon. Nevertheless, this issue can be readily resolved by manually identifying and

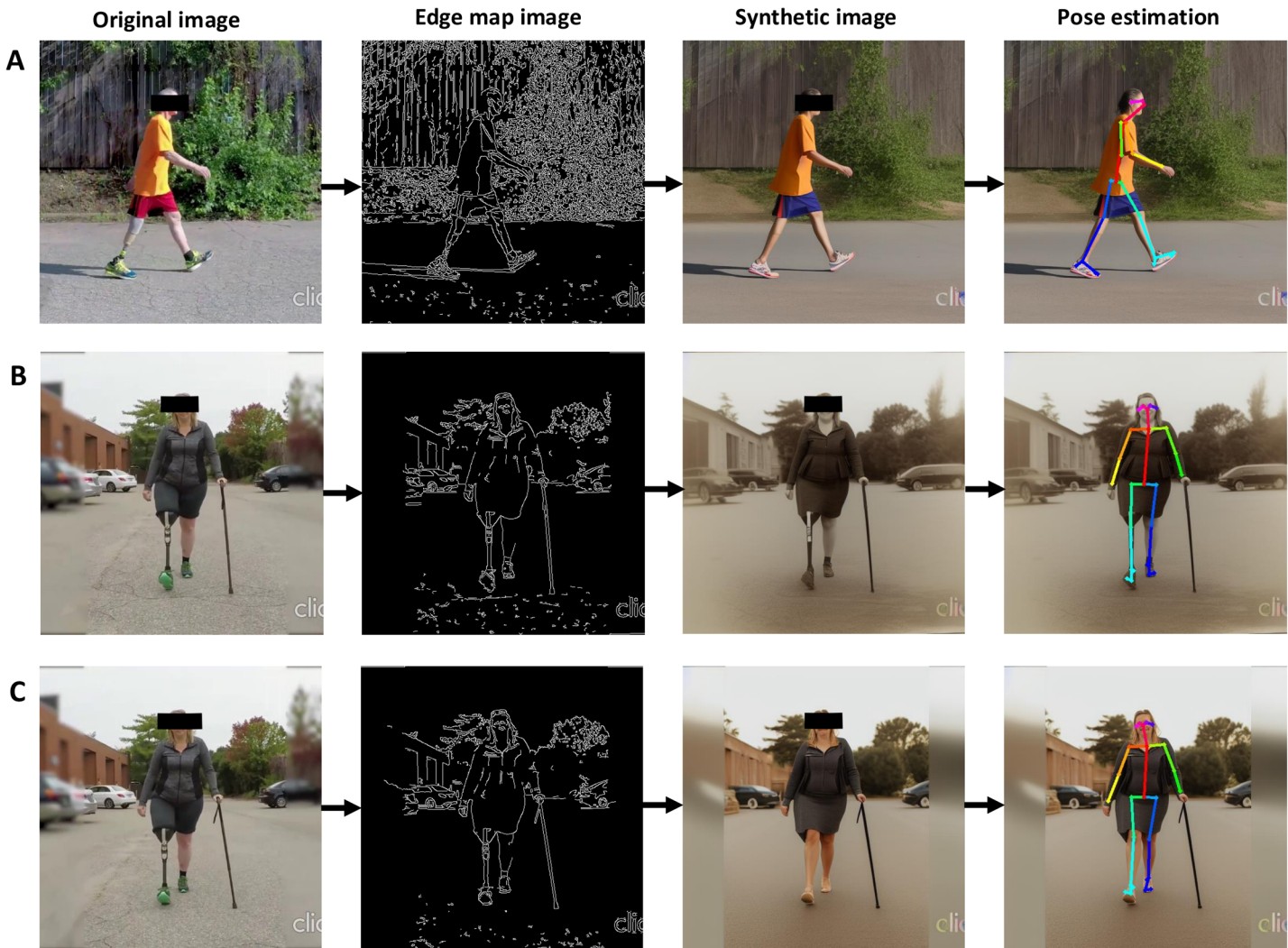

**Fig 6. Different case examples.** (A) In the original image, the left limb is positioned in front. However, upon generating the synthetic image, the left leg was positioned backward, and this discrepancy persisted after the application of OpenPose, where the dark blue line represents the left leg. (B) Case example where the generation of the synthetic image fails to change the prosthetic limb to an intact limb. (C). Case example where the generation of the synthetic image successfully changes the prosthetic limb to an intact limb.

correcting the swapped frames or by utilizing other cues, such as the asynchronous movement of the ipsilateral arm and leg swing. This observation is reinforced by the finding that, although the zero-shot method requires a higher number of frames to be swapped compared to OpenPose, it still leads to a decrease in error. It is worth highlighting that the default Open-Pose on the original image may also cause the lower-limb keypoints to swap, similar to previous studies [24,42]. Moreover, the zero-shot method resulted in the reduction of frames where keypoints were not detected in OpenPose due to the non-resemblance of intact limbs (S1 Table). This further demonstrates the efficacy of the zero-shot method in converting the prosthetic limb to a human-like representation, thereby allowing the application of pose estimation.

It is important to acknowledge that not all frames can be successfully transformed into images resembling able-bodied individuals (Fig 6B) due to the limitations of edge map detection.

Nevertheless, even when comparing frames at approximately the same gait cycle, the generation of synthetic images yields inconsistent results (Fig 6C). This inconsistency is more pronounced in the case of transfemoral amputees, likely attributed to the dissimilarity between the prosthetic limb and a human limb, posing challenges for the model in identifying the appropriate image type. Despite inputs of prompts into ControlNet, emphasizing the presence of two intact limbs, the model may still encounter difficulties. However, it is worth noting that the majority of frames can still be successfully transformed, as evidenced by the overall reduction in coordinate error, especially for the prosthetic limb. Another limitation pertains to the wide range of prosthetic options available in the market. Our testing of the diffusion model was limited to a specific subset of prosthetic users, which may introduce some variability in the results. Nevertheless, we anticipate that any deviations from our findings will be minor and not substantially different.

The main advantage of the proposed method is its ability to perform zero-shot pose estimation for prosthetic users without the need for any manual labelling and model training. However, the zero-shot ability comes with the limitations of slow processing speed due to the need to run inference on the large image generative diffusion model. Thus, there is a trade-off between inference time versus the time required for manual labelling for prosthetic joints and finetuning or training a custom detection model. With personal computing hardware (Nvidia GTX Titan X), the speed of generating 512 x 512 synthetic images is approximately 8 - 10 seconds per image which presents the main bottleneck for the workflow. This means for a 24 FPS video, the inference time needed is 200x that of the video length. Given a walking speed of 1 m/s, a 10 m walk test may require 10 seconds of video recording and may take more than 30 minutes to process which precludes real-time analysis and presents limitations for practical clinical usage.

To address this, we propose 2 methods to speed up inference. First, clinicians may run pose estimation on the original video to identify poorly estimated frames before employing the image-generative zero-shot workflow on the poorly estimated frames only. Second, clinicians may drop selected frames to reduce the number of images to be generated. For the purpose of gait analysis, the gait cycle is obtained by averaging over many gaits. In such use cases, it is possible to remove selected frames and perform interpolation without affecting the resultant gait cycle significantly. Nevertheless, the zero-shot method is still faster than creating a custom-trained image set which includes manual annotations and a longer training time for custom pose estimation models for each individual.

Another potential alternative for regenerating an intact limb image for lower-limb amputees is inpainting a masked section of the image with diffusion models [43]. This technique involves selectively highlighting specific parts of an image that require modification. However, this option was not explored in our study due to the continuous movement of the limbs in each image, which necessitates manual masking and thereby increases post-processing time. Future research could investigate alternative methods for automating the masking of the prosthetic limb in each frame. By generating images for smaller masked regions, it may be possible to expedite the inference time.

It is important to note that our ground truth keypoints were obtained using DeepLab-Cut. While DeepLabCut is a powerful tool for pose estimation, it may introduce some uncertainties and potential errors. To enhance the robustness of our findings, further validation using traditional marker-based motion capture systems would be beneficial. Comparing the results of our zero-shot method with data from a marker-based system would provide

a more comprehensive evaluation of its performance. This additional validation step could strengthen the credibility of our approach and offer valuable insights into any discrepancies between markerless and marker-based techniques, particularly in the context of prosthetic gait analysis.

The use of YouTube videos as our data source introduces its own set of limitations. These videos vary in quality and recording conditions can affect the performance of our method. Additionally, the calculation of the MAE involves an assumption about the subject's height, potentially introducing inaccuracies. While the results from these open-source videos are promising, practitioners aiming to replicate our method may benefit from controlled video recordings with known parameters, potentially leading to improved outcomes. For instance, optimizing background conditions could enhance edge detection quality and subsequent pose estimation. However, this study also demonstrates that despite uncontrolled recording conditions, valuable information can still be extracted, making the method applicable to existing videos.

In addition to these technical limitations, it is crucial to acknowledge the constraints of our data set. We assessed the proposed method on only two individuals. As such, our results should be considered as a preliminary report and proof of concept, demonstrating that the zero-shot method is a potential solution for obtaining kinematics data from lower-limb amputees. The zero-shot performance may vary across a broader population with diverse prosthetic designs and individual characteristics. Further research with a larger and more diverse sample size is necessary to validate the robustness and wide applicability of this zero-shot method.

## Supporting information

**S1 Fig. A schematic diagram representing the breakdown of videos for each type of amputation.**
(PDF)

**S1 Table. Number of frames with issues in each camera view based on each method.**
(XLSX)

**S2 Table. Individual keypoints coordinates error (MAE) when comparing DeepLabCut (treated as ground truth) and OpenPose.**
(XLSX)

**S3 Table. Joint angle error (MAE) when comparing DeepLabCut (ground truth) and OpenPose.**
(XLSX)

**S4 Table. Review of literature that has assessed the accuracy of OpenPose on gait analysis.**
(XLSX)

**S5 Table. Individual keypoints coordinates error (MAE) when comparing manual annotation and the different pose estimations.** It is established here that DeepLabCut is satisfactory to be used as ground truth labels.
(XLSX)

**S6 Table. Joint angle error (MAE) when comparing manual annotation and the different pose estimations.**
(XLSX)

**S7 Table. Number of frames where the likelihood is less than 0.50.**
(XLSX)

## Author contributions

**Conceptualization:** Tianxun Zhou, Keng-Hwee Chiam.

**Data curation:** Tianxun Zhou, Muhammad Nur Shahril Iskandar.

**Formal analysis:** Muhammad Nur Shahril Iskandar.

**Investigation:** Tianxun Zhou, Muhammad Nur Shahril Iskandar.

**Methodology:** Tianxun Zhou, Muhammad Nur Shahril Iskandar.

**Project administration:** Keng-Hwee Chiam.

**Supervision:** Keng-Hwee Chiam.

**Visualization:** Tianxun Zhou, Muhammad Nur Shahril Iskandar.

**Writing – original draft:** Tianxun Zhou, Muhammad Nur Shahril Iskandar.

**Writing – review & editing:** Tianxun Zhou, Muhammad Nur Shahril Iskandar, Keng-Hwee Chiam.

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
