## [Decision Letter · Decision Letter 0]

3 Sep 2024

PDIG-D-24-00176

Diffusion models enable zero-shot pose estimation for lower-limb prosthetic users

PLOS Digital Health

Dear Dr. Chiam,

Thank you for submitting your manuscript to PLOS Digital Health. After careful consideration, we feel that it has merit but does not fully meet PLOS Digital Health's publication criteria as it currently stands. Therefore, we invite you to submit a revised version of the manuscript that addresses the points raised during the review process.

Please submit your revised manuscript within 60 days Nov 02 2024 11:59PM. If you will need more time than this to complete your revisions, please reply to this message or contact the journal office at digitalhealth@plos.org. Please include the following items when submitting your revised manuscript:

We look forward to receiving your revised manuscript.

PLOS Digital Health

Editorial Team 

Journal Requirements:

1.Please ensure that you provide a single, cohesive .tex source file for your LaTeX revision. You may upload this file as the item type 'LaTeX Source File.' As stated in the PLOS template, your references should be included in your .tex file (not submitted separately as .bib or .bbl). Please also ensure that you are making any formatting changes to both your .tex file and the PDF of your manuscript. If you have any questions, please contact Latex@plos.org. You can find our LaTeX guidelines here:

https://journals.plos.org/digitalhealth/s/latex

**

Reviewers' comments:

Reviewer's Responses to Questions

**Comments to the Author**

1. Does this manuscript meet PLOS Digital Health’s publication criteria? Is the manuscript technically sound, and do the data support the conclusions? The manuscript must describe methodologically and ethically rigorous research with conclusions that are appropriately drawn based on the data presented.

Reviewer #1: Yes

Reviewer #2: Partly

Reviewer #3: Yes

2. Has the statistical analysis been performed appropriately and rigorously?

Reviewer #1: Yes

Reviewer #2: Yes

Reviewer #3: Yes

3. Have the authors made all data underlying the findings in their manuscript fully available (please refer to the Data Availability Statement at the start of the manuscript PDF file)?

Reviewer #1: Yes

Reviewer #2: No

Reviewer #3: Yes

4. Is the manuscript presented in an intelligible fashion and written in standard English?

PLOS Digital Health does not copyedit accepted manuscripts, so the language in submitted articles must be clear, correct, and unambiguous. Any typographical or grammatical errors should be corrected at revision, so please note any specific errors here.

Reviewer #1: Yes

Reviewer #2: Yes

Reviewer #3: Yes

5. Review Comments to the Author

Please use the space provided to explain your answers to the questions above. You may also include additional comments for the author, including concerns about dual publication, research ethics, or publication ethics. (Please upload your review as an attachment if it exceeds 20,000 characters)

Reviewer #1: The paper titled "Diffusion models enable zero-shot pose estimation for lower-limb prosthetic users" presents a novel approach for improving pose estimation for lower-limb prosthetic users using generative diffusion models. This approach transforms images of prosthetic limbs into able-bodied representations, allowing standard pose estimation models to better detect key points without requiring additional data collection or model retraining.

The use of diffusion models to transform prosthetic limb images into able-bodied representations is a creative solution to a challenging problem. Additionally the method shows substantial improvements with reductions in coordinate errors of 37% for transtibial and 76% for transfemoral prosthetic limbs. However some minor modifications are suggested for the publication:

- The authors should verify clarity and writing, to be more concise, for example in introduction lines 43 - 50 are really similar to 51 - 59.

- the method is clear in text, but improving the figure 6 and adding more details could help to clarify the methodology used.

- More details on the edge map generation process and its potential limitations would be helpful. The authors should discuss how variations in edge map quality might affect the overall pose estimation accuracy.

- The specific configuration and parameters used in ControlNet or to the data should be elaborated on, including any modifications made for this study. 

- The use of publicly available videos from YouTube introduces variability in video quality and recording conditions. The paper should discuss the implications of these constraints on the generalizability of the results.

- The study relies on DeepLabCut for ground truth keypoints, but further validation with more robust ground truth data would strengthen the findings. It would be beneficial to include more detailed comparisons with other ground truth datasets if available.

- The results section is thorough, but could benefit from a clearer separation of key findings and their implications. Highlighting the most significant results in a summary table or bullet points could enhance clarity.

Reviewer #2: This study by Zhou et al. presents a new method for video-based gait analysis in prothetic users that corrects errors in keypoint-tracking of the prosthetic limb. This is important because a conventional and pre-trained pose estimation algorithm like OpenPose does not track the prosthetic limb well. The authors find that this new “zero-shot” method improves lower-limb keypoint tracking of the prosthetic limb compared with OpenPose. 

There is no doubt that this paper addresses an important issue: video-based gait analysis of prosthetic gait is an important topic and that there are challenges with correctly tracking the movements of the prosthetic limb. However, I have concern about the limited data set used for validation in the study. Considering that only 4 unique gait trials were used of 2 individuals – I think the manuscript should be considered a preliminary report which must be clearly highligted. 

Major Comments 

1) The data set only comprises 2 individuals and only 4 unique gait trials (captured by 4 cameras giving a total of 16 videos). Furthermore, it appears that only the sagittal videos were used which would bring the number of videos down to 8. And last, only the limb closest to the camera was analyzed. From an already limited data set, this whittles down the data set significantly. 

While I have no doubt that the “zero-shot” method dramatically improves keypoint tracking relative to OpenPose – this paper is really more of a preliminary report. I’d like the authors to consider sourcing additional videos for this paper. Alternatively, data limitation must be highlighted in the abstract, results and conclusion. 

2) I’d like the authors to reflect on the reliability of the ground truth keypoint tracking using DLC. First, how are keypoints defined at the “knee”, “ankle” and “toes” in a prosthetic limb? Second, the authors report an MAE of 8 pixels using DLC (line 411). This translates to an error of ~2.4 cm (using conversion on line 140). Is 2.4 cm within an acceptable limit of joint location in a prosthetic limb? 

Minor Comments 

1) Introduction: The intro is quite long and I can it will be improved by omitting the section on observational gait analysis which is outside the scope of the paper. 

2) Table 1: Why are MAE of prosthetic transfemoral joint coordinates using OpenPose reported as >100? 

3) Methods, line 419: How many frames with likehood less than 0.50 were removed? Please report in manuscript. 

4) Methods, lines 437-441: lateral trunk lean is described here, but is not reported in the Results. 

5) Figures 2-4: The is repetition across the figures as curves are duplicated between 2 and 3 and between 3 and 4. I strongly suggest that the authors either collapse into a single figure of many subplots or divide into 2 figures. 

6) Figures 5 and 6: the image quality is very poor.

Reviewer #3: This manuscript meets the PLOS Digital Health’s publication criteria in terms of:

1. Originality - a novel zero-shot pose estimation method to achieve accurate pose estimation for lower limb prosthetic users is presented;

2. High Importance and broad interest - the significant problem of markerless pose estimation models in accurate localization of joints on prosthetic limbs for quantitative gait analysis is outermost important for assessing and rehabilitating lower-limb amputee population;

3. High methodological rigor and ethical standards - methods are robust and appropriate for the research question and ethical standards are maintained since publicly accessible videos from a non-profit organization were used;

4. Substantial evidence for its conclusions - conclusions are supported by convincing results following a thorough and appropriate analysis, complemented with supplementary material;

5. Clearly outlined utility and accessibility - findings are straightforward presented to be easily understood and utilized by the broader digital health community;

6. Appropriate standards and practice of open science - the manuscript adhere to principles of transparency reproducibility (methods described in detail), and data sharing (data available in the supplementary material and reference to the GitHub code for the adopted model).

The methods used for statistical evaluation are clearly described, and appropriate techniques (with reference to current literature methods) have been employed to ensure the robustness and validity of the results. Statistical tests on the comparison (e.g., in terms of normalized joint angle kinematics curves) of the zero-shot pose estimation method to ground truth and standard pose estimation models (i.e., OpenPose) can be added to enrich the statistical analysis.

As the authors mention, the raw data supporting the conclusions of the manuscript need to be made available. 

The text is clear and well-structured, allowing readers to easily understand the research objectives, methods, results, and conclusions. However, it’s advised to move the Method section before the Results paragraph for a more comprehensive outline of the study’s findings. 

The use of standard English throughout the manuscript ensures that it is accessible to a broad audience. 

Finally, to further improve the manuscript, comments in the document uploaded by the reviewer should be addressed.

6. PLOS authors have the option to publish the peer review history of their article (what does this mean?). If published, this will include your full peer review and any attached files.

**Do you want your identity to be public for this peer review?** For information about this choice, including consent withdrawal, please see our Privacy Policy.

Reviewer #1: Yes: David Restrepo

Reviewer #2: No

Reviewer #3: No

---

## [Decision Letter · Decision Letter 1]

8 Jan 2025

Diffusion models enable zero-shot pose estimation for lower-limb prosthetic users

PDIG-D-24-00176R1

Dear Dr. Chiam,

We are pleased to inform you that your manuscript 'Diffusion models enable zero-shot pose estimation for lower-limb prosthetic users' has been provisionally accepted for publication in PLOS Digital Health.

Best regards,

Chiara Corti, MD

Academic Editor

PLOS Digital Health

**Additional Editor Comments (if provided):**

The authors addressed the reviewers' comments.

I only minor points in wording or style that I would like the authors to address:

-Page 2, line 8: please change to “physiological differences” (plural)

-Page 2, line 9-10: what is the “increase in O2 consumption of ~49%” in relation to?

-Page 2, line 26: please write the names of the authors of ref 15

-Page 2, line 37: replace “arises” with “has emerged”

-Page 3, line 66: please change to “lower-limb prosthetics” (plural)

-Page 3, line 75: please write the names of the authors of ref 30

-Page 3, line 78: change “include” to “including”

-Page 4, line 102: change “for example” to “specifically”

-Page 5, line 161: what are “3M images”?

**Reviewer Comments (if any, and for reference):**

Reviewer's Responses to Questions

**Comments to the Author**

1. If the authors have adequately addressed your comments raised in a previous round of review and you feel that this manuscript is now acceptable for publication, you may indicate that here to bypass the “Comments to the Author” section, enter your conflict of interest statement in the “Confidential to Editor” section, and submit your "Accept" recommendation.

Reviewer #2: (No Response)

Reviewer #3: All comments have been addressed

Reviewer #4: All comments have been addressed

2. Does this manuscript meet PLOS Digital Health’s publication criteria? Is the manuscript technically sound, and do the data support the conclusions? The manuscript must describe methodologically and ethically rigorous research with conclusions that are appropriately drawn based on the data presented.

Reviewer #2: Yes

Reviewer #3: Yes

Reviewer #4: Yes

3. Has the statistical analysis been performed appropriately and rigorously?

Reviewer #2: Yes

Reviewer #3: Yes

Reviewer #4: N/A

4. Have the authors made all data underlying the findings in their manuscript fully available (please refer to the Data Availability Statement at the start of the manuscript PDF file)?

Reviewer #2: No

Reviewer #3: Yes

Reviewer #4: Yes

5. Is the manuscript presented in an intelligible fashion and written in standard English?

Reviewer #2: Yes

Reviewer #3: Yes

Reviewer #4: Yes

6. Review Comments to the Author

Reviewer #2: I thank the authors for adequately addressing my comments.

I only minor points in wording or style that I would like the authors to address:

Page 2, line 8: please change to “physiological differences” (plural)

Page 2, line 9-10: what is the “increase in O2 consumption of ~49%” in relation to?

Page 2, line 26: please write the names of the authors of ref 15

Page 2, line 37: replace “arises” with “has emerged”

Page 3, line 66: please change to “lower-limb prosthetics” (plural)

Page 3, line 75: please write the names of the authors of ref 30

Page 3, line 78: change “include” to “including”

Page 4, line 102: change “for example” to “specifically”

Page 5, line 161: what are “3M images”?

Page 5, line 178: stating “we introduce OpenPose in more detail” sounds like you are modifying OpenPose and introduces a modified pose estimation algorithm. I would replace with “we summarize OpenPose”

Reviewer #3: (No Response)

Reviewer #4: In general, you have described a good idea in current article. Adding a short conclusion in the end would be beneficial. As for the future, it would be interesting to test your concept on the bigger cohort of patients

7. PLOS authors have the option to publish the peer review history of their article (what does this mean?). If published, this will include your full peer review and any attached files.

**Do you want your identity to be public for this peer review?** For information about this choice, including consent withdrawal, please see our Privacy Policy.

Reviewer #2: No

Reviewer #3: No

Reviewer #4: No
